# Novel Model for Stomatal Conductance: Enhanced Accuracy Under Variable Irradiance and CO_2_ in C_3_ Plant Species

**DOI:** 10.3390/biology14111501

**Published:** 2025-10-27

**Authors:** Zipiao Ye, Ting An, Xiaolong Yang, Huajing Kang, Fubiao Wang

**Affiliations:** 1New Quality Productivity Research Center, Guangdong ATV College of Performing Arts, Zhaoqing 526631, China; yezp@jgsu.edu.cn (Z.Y.); anting_6918@163.com (T.A.); 2Institute of Biophysics, Math & Physics College, Jinggangshan University, Ji’an 343009, China; 3Medical College, Guangdong ATV College of Performing Arts, Zhaoqing 526631, China; 4School of Life Sciences, Nantong University, Nantong 226019, China; yangxl@ntu.edu.cn; 5State Key Laboratory of Environmental Chemistry and Ecotoxicology, Research Center for Eco-Environmental Sciences, Chinese Academy of Sciences, Beijing 100085, China; 6Wenzhou Academy of Agricultural Sciences, Wenzhou 325006, China; 7Wenzhou Key Laboratory of Agricultural & Forestry Carbon Sequestration and Tea Resource Development, Wenzhou 325006, China

**Keywords:** stomatal conductance to CO_2_, Ball-Woodrow-Berry model, Medlyn model, light-response to photosynthesis, CO_2_-response to photosynthesis

## Abstract

**Simple Summary:**

This study introduces a new model for predicting stomatal conductance (*g*_sc_)—the rate at which plants exchange gas such as CO_2_ and water vapor with the atmosphere through stomata—in three common C_3_ plant species. The researchers compared a new model proposed by Ye et al. with two widely used models (Ball–Woodrow–Berry and Medlyn models) under varying light intensities and CO_2_ conditions. The study found that the new model more accurately describes how *g*_sc_ responds to changes in the environment, especially under high light or fluctuating CO_2_ levels. The results highlight the limitations of existing models and demonstrate the improved predictive power of the new approach. This work is important for better understanding plant water use efficiency, improving crop productivity, and predicting how plants may respond to climate change.

**Abstract:**

This study analyzes stomatal conductance (*g*_sc_) in *Trifolium repens* L., *Lolium perenne* L., and *Triticum aestivum* L. under varying environmental conditions. Light-response curves for photosynthesis (*A*_n_–*I*) at 420 μmol mol^−1^ CO_2_ were used to determine saturating irradiance (*I*_sat_) using a light-response model for photosynthesis, and CO_2_-response curves for photosynthesis (*A*_n_–*C*_i_) were measured at *I*_sat_ and half *I*_sat_ for these C_3_ plant species. The Ball–Woodrow–Berry (BWB) model, Medlyn model, and a new model were compared for their ability to describe the net photosynthetic rate (*A*_n_) relative to *g*_sc_ under changing irradiance or CO_2_. The BWB model overestimated *g*_sc_ response, simplifying stomatal behavior, while the Medlyn model deviated at high *A*_n_ values, indicating limitations in dynamic responses. The new model showed a better empirical fit under the tested conditions, achieving high *R*^2^ values and low AIC values across all three species, and demonstrated a strong alignment with empirical data. Our findings highlight the complexity of *g*_sc_ regulation and the need for improved models to better represent stomatal dynamics under different environmental conditions. This research is vital for optimizing water use efficiency, enhancing crop productivity, and understanding plant resilience to climate change.

## 1. Introduction

Stomatal conductance (*g*_s_; abbreviations listed in Appendix A) is a critical physiological parameter in plant biology, playing a central role in regulating gas exchange between the plant and the atmosphere [1,2]. Specifically, stomatal conductance refers to the rate at which water vapor and carbon dioxide move through the stomatal pores, which are microscopic openings on the surfaces of plant leaves. This regulation is essential for maintaining plant health and productivity, as it directly influences key processes such as photosynthesis, transpiration, and the plant’s overall ability to cope with environmental stressors [3,4,5].

The dynamics of stomatal opening and closing are influenced by a range of environmental factors, including light intensity, atmospheric carbon dioxide concentration, humidity, and soil water availability [6,7,8,9,10]. These responses are part of a complex feedback mechanism that enables plants to optimize gas exchange. Through this process, plants balance the need to absorb carbon dioxide for photosynthesis with the risk of water loss due to transpiration. This delicate balance is crucial for plant survival, particularly in the face of changing environmental conditions such as drought or fluctuating temperatures [5,11,12].

Historically, stomatal conductance has been modeled using a variety of approaches, ranging from simple empirical relationships to more mechanistic models that account for the underlying physiological processes [1,2,13,14,15,16,17]. One of the most widely used models is the Ball–Woodrow–Berry (BWB) model, which provides a framework for understanding how stomatal conductance is influenced by atmospheric demand for water vapor and the plant’s internal water status [18]. This model has been instrumental in understanding the coupling between stomatal conductance and transpiration, but it does not fully account for the complex interactions between stomatal behavior and other environmental variables [19]. Based on the BWB model, other models of stomatal conductance were developed [1,11,19,20,21,22].

As research in plant physiology has advanced, there is an increasing recognition that more sophisticated, process-based models are necessary to accurately predict stomatal behavior across a wide range of environmental conditions. These models seek to integrate multiple physiological processes, including the regulation of stomatal aperture by both biochemical and biophysical factors, and can help predict plant responses to climate change and water stress [3,8]. For instance, models that incorporate mechanistic descriptions of stomatal signaling pathways, such as those proposed by Ainsworth and Rogers [23], provide a more comprehensive understanding of how plants adjust stomatal conductance in response to both internal and external cues.

In recent years, advancements in computational plant modeling, along with the development of high-throughput phenotyping techniques, have significantly improved our ability to predict stomatal responses in dynamic and complex environments [17,24,25]. These cutting-edge models are not only essential for advancing fundamental research in plant physiology but also have important practical implications, especially for agriculture. Optimizing water use efficiency, enhancing crop productivity, and understanding plant resilience to climate change are key areas where accurate predictions of stomatal conductance can have a direct impact [26]. Currently, while the stomatal conductance model developed by Medlyn et al. [26] enjoys widespread adoption, it is not without its recognized challenges and constraints. Research findings underscore the model’s difficulty in precisely forecasting stomatal responses to fluctuations in atmospheric CO_2_ levels (*C*_a_). Specifically, under conditions where carboxylation is the limiting factor—characterized by intense light and scant CO_2_—the model erroneously anticipates a stomatal opening, contrary to the observed stomatal closure [27]. Concurrently, the model falls short in comprehensively capturing the structural acclimation of plants to enduring CO_2_ variations, potentially leading to inaccuracies in parameters such as maximum hydraulic conductance and photosynthetic capacity adjusted for temperature [27]. Moreover, the Medlyn model [26], which is grounded in the optimization theory of water-use efficiency, transitions to an empirically sound yet suboptimal framework under conditions of light saturation, potentially failing to accurately mirror the physiological dynamics of stomata [27]. These limitations underscore an imperative for ongoing research and model enhancement to deepen our understanding and predictive capabilities regarding stomatal behavior across diverse environmental scenarios. Furthermore, in various stomatal conductance models [18,20,21,22,26], the term *g*_0_ refers to the residual stomatal conductance, which is generally assumed to approach zero [2,28,29]. However, the extent to which this assumption holds true in actual conditions remains uncertain, warranting further investigation into this matter.

The objectives of this study are: (i) to plot *A*_n_–*C*_a_ and *A*_n_–*g*_sc_ response curves for *Trifolium repens* L., *Lolium perenne* L., and *Triticum aestivum* L under both saturating irradiance (*I*_sat_) and half *I*_sat_, thereby elucidating the response patterns of *A*_n_ to *g*_sc_ under these light conditions; (ii) to compare the ability of the novel model proposed by Ye et al. [2,30] with the BWB and Medlyn models in describing the relationship between *A*_n_ and *g*_sc_ under conditions of varying irradiance or CO_2_ concentration; and (iii) to analyze the rationality of considering the *g*_0_ values obtained from different model’s fittings as the conventionally defined residual stomatal conductance, or whether they are small enough to be negligible, by comparing the *g*_0_ values derived from the different models. This comprehensive approach will not only allow us to evaluate the effectiveness of these models but also deepen our understanding of the physiological mechanisms governing stomatal conductance in response to environmental variations.

## 2. Materials and Methods

### 2.1. Stomatal Conductance Models Description

#### 2.1.1. Stomatal Conductance of Ball–Woodrow–Berry Model

The Ball–Woodrow–Berry (BWB) model is an empirical construct that captures the dynamic relationship between stomatal conductance and the net photosynthetic rate (*A*_n_). Originally developed by Ball et al. [18], this model characterizes stomatal conductance as a function of *A*_n_, leaf surface CO_2_ concentration (*C*_s_), and relative humidity (*h*_r_). It posits that under steady-state conditions, stomatal conductance exhibits a linear correlation with *A*_n_, provided that *C*_s_ and *h*_r_ remain constant. This relationship is predicated on the stomata’s role in modulating intercellular CO_2_ concentration to foster a stable environment for photosynthesis. The stomatal conductance to CO_2_ (*g*_sc_) is determined using the Ball–Woodrow–Berry model [18]:(1)$$g_{\mathrm{sc}}=g_{1}\frac{A_{n}h_{r}}{C_{s}}+g_{0}$$
where *g*_0_ (uint: mol·m^−2^·s^−1^) and *g*_1_ (dimensionless) are the fitting parameters [2,18,31], *A*_n_ is net photosynthetic rate (μmol·m^−2^·s^−1^), *h*_r_ is relative humidity at the leaf surface (dimensionless), and *C*_s_ is atmospheric CO_2_ concentration at the leaf surface. The parameter *g*_1_ represents the slope of the relationship between *g*_sc_ and *A*_n_*h*_r_/*C*_s_ [26] or is referred to as the empirical slope [32].

However, in mathematical contexts, the ratio of two large numbers is often highly sensitive to minor changes in the denominator. To mitigate this sensitivity, Equation (1) can be reformulated as:(2)$$A_{n}=\frac{1}{g_{1}h_{r}}\left(g_{\mathrm{sc}}-g_{0}\right)C_{s}$$

Therefore, the parameters *g*_1_ and *g*_0_ can be determined by applying Equation (2). Furthermore, in Equation (2), when *A*_n_ is equal to zero, it follows that *g*_sc_ must equal *g*_0_.

#### 2.1.2. Stomatal Conductance Model of Medlyn et al. [26]

The optimal stomatal conductance model presented by Medlyn et al. [26] is defined by the following equations:(3)$$g_{\mathrm{sc}}=g_{0}+\left(1+\frac{g_{1}}{\sqrt{D}}\right)\frac{A_{n}}{C_{s}}\;$$(4)$$g_{1}\propto \sqrt{\Gamma_{*}\lambda }$$
where *g*_0_ signifies the stomatal conductance when the net photosynthetic rate is zero (0 μmol⋅m^−2^⋅s^−1^), *g*_1_ represents the slope of the stomatal conductance response, *λ* is the marginal water cost of carbon gain, and *Γ*_*_ denotes the CO_2_ compensation point concentration in the absence of day respiratory rate (*R*_day_). The stomatal conductance slope *g*_1_ is proportional to the combination of terms $\sqrt{\Gamma_{*}\lambda }$ [26], suggesting that *g*_1_ rises with increasing *λ* values. Similarly to Equation (2), the third equation can be rewritten as:(5)$$A_{n}=\frac{1}{\left(1+\frac{g_{1}}{\sqrt{D}}\right)}\left(g_{\mathrm{sc}}-g_{0}\right)C_{s}$$

Hence, the parameters *g*_1_ and *g*_0_ can be determined by applying Equation (5). Consistent with Equation (2), in Equation (5), when *A*_n_ is zero, it is necessary that *g*_sc_ equals *g*_0_.

#### 2.1.3. Stomatal Conductance Model by Ye et al. [2]

The stomatal conductance model of Ye et al. [2] is expressed as:(6)$$g_{\mathrm{sc}}=g_{1}\frac{A_{n}}{C_{a}-C_{i}}+g_{0}$$
where *C*_i_ is the intercellular CO_2_ concentration, *g*_1_ is a coefficient (dimensionless).

However, to address the sensitivity of large number ratios and to align more closely with the mathematical expression of Fick’s first law of diffusion, Equation (6) can be reformulated as follows:(7)$$A_{n}=\frac{1}{g_{1}}\left(g_{\mathrm{sc}}-g_{0}\right)\left(C_{a}-C_{i}\right)$$

Equation (7) elucidates that *A*_n_ is directly proportional to the difference between *C*_a_ and *C*_i_, and it exhibits a linear relationship with *g*_sc_ under specific environmental conditions. The parameters *g*_1_ and *g*_0_ can be determined by employing Equation (7) to model the response of *A*_n_ to variations in *C*_a_, *C*_i_ and *g*_sc_. Moreover, in Equation (7), when *A*_n_ equals zero, it follows that either *g*_sc_ equals *g*_0_ or *C*_a_ equals *C*_i_.

Additionally, in Equation (7), when *g*_1_ is set to 1 and *g*_0_ to 0, the equation reduces to the form of Fick’s first law of diffusion. Following the principles of Fick’s first law, under steady-state conditions, the *A*_n_ can be articulated as detailed below [9,33,34,35]:(8)$$A_{n}=g_{\mathrm{sc}}\left(C_{a}-C_{i}\right)$$

### 2.2. Study Site and Plants

*T. repens*, *L. perenne* (Zhongxin 830) and *T. aestivum* (Jimai 22) were used as test materials. The experimental site was located at Yucheng Station in the southwest of Yucheng County, Chinese Academy of Sciences, Shandong Province. The plants were sown on 15 October 2022, and maintained under standard field conditions. Data collection was performed on sunny days from 28 April to 10 May 2023. *T. repens* was in a period of the BBCH 14–16 (Biologische Bundesanstalt, Bundessortenamt und CHemical Industry, Julius Kühn-Institut (JKI), Quedlinburg, Germany) growth stage (vigorous vegetative stage with 4–6 leaves unfolded), with an approximate plant height of 15 cm, and mature leaves at the top were selected for testing. *L. perenne* was in the BBCH 45–49 growth stage (booting stage), with an approximate plant height of 1.3 m. The new mature, fully unfolded leaves were selected for testing. *T. aestivum* was in the BBCH 45–55 growth stage (booting to early flowering stage), with an approximate plant height of 60–70 cm, and the flag leaf was selected for testing.

### 2.3. Gas Exchange and Chl Fluorescence Measurement

From 28 April to 10 May 2023, *A*_n_–*I* curves were measured on each study plant with an open-path gas exchange system (LI-6400; Li-Cor, Lincoln, NE, USA), equipped with a leaf chamber fluorometer (6400-40; Li-Cor). Measurements were conducted between 9:30–11:30 and 14:30–17:00 on sunny days. The ambient [CO_2_] concentration was maintained at 420 μmol·mol^−1^ for 15 or 13 light levels in the following order (first to last): 2000, 1800, 1600, 1400, 1200, 1000, 800, 600, 400, 200, 150, 100, 80, 50, 0 μmol·m^−2^·s^−1^ for *L. perenne* and *T. aestivum*; and 1600, 1400, 1200, 1000, 800, 600, 400, 200, 150, 100, 80, 50, 0 μmol·m^−2^·s^−1^ for *T. repens*. The plants were allowed to acclimate to changes in light intensity for approximately 2–3 min before measurements were logged; it took about 50 min to complete an entire *A*_n_–*I* curve. After data collection, a mechanistic model of *A*_n_–*I* in “Photosynthesis Model Simulation Software (PMSS version 2.0)” platform was used to simulate the *A*_n_–*I* curves [36], This simulation determined the saturating irradiance (*I*_sat_) as 1200 μmol·m^−2^·s^−1^ for *T. repens*, 900 μmol·m^−2^·s^−1^ for *L. perenne*, and 2000 μmol·m^−2^·s^−1^ for *T. aestivum*, respectively. Then *A*_n_–*C*_i_ and *J*–*C*_i_ curves were simultaneously recorded at saturating irradiance for 12 CO_2_ concentrations in the following order (first to last measurement): 1600 (except for *T. aestivum*), 1400, 1200, 1000, 800, 600, 420, 300, 200, 100, 60, and 0 μmol·mol^−1^. The measurements were conducted using an open-path gas exchange system. A stable injected gas supply was provided by CO_2_ cartridges (iSi Gesellschaft mit beschränkter Haftung (GmbH), AUSTRIA U2614, Vienna, Austria) compatible with the LI-6400, coupled with the instrument’s built-in CO_2_ mixer. The flow rate entering the leaf chamber was set and maintained at 400 μmol s^−1^. The temperature within the leaf chamber was set at 30 ± 1.3 °C, and the relative humidity was controlled between 45% and 75%. To ensure steady-state conditions, the plants were given approximately 5 min to acclimate to ambient CO_2_ (420 μmol·mol^−1^) in the gas exchange chamber before beginning each *A*_n_–*C*_i_ and *J*–*C*_i_ curve, and then logged. It took approximately 45 min to complete a single *A*_n_–*C*_i_ and *J*–*C*_i_ curve. The gas exchange properties including photosynthesis rate, *C*_i_, leaf temperature and stomatal conductance were logged at each *C*_a_ once the system had reached a predetermined stability point (coefficient of variation ≤ 1%). If this criterion could not be met within a reasonable waiting period (typically 3–5 min), manual logging was performed once the readings were confirmed to have stabilized. Gas exchange measurements were conducted on three independent biological replicates for each plant species (*n* = 3). The three models of stomatal conductance (BWB model, Medlyn model and a new model proposed by Ye et al.) were integrated into the Photosynthesis Model Simulation Software (PMSS) platform (http://www.zipiao.tech, accessed on 13 November 2024) (Zipiao software development Co., Ltd., Ji’an China), which was then used to simulate these data to obtain *g*_0_ and *g*_1_ for the three C_3_ plants. The parameters *g*_0_ and *g*_1_ were estimated using nonlinear least squares regression implemented in the “Photosynthesis Model Simulation Software (PMSS)” platform.

### 2.4. Statistical Analysis

All variables are presented as mean values (±*SE*) based on three replicate samples for each species. The data were subjected to one-way analysis of variance (ANOVA). To determine if there were significant differences between *A*_n_ values calculated using Equations (2), (5) and (7), and the corresponding observed values, we employed a paired-sample *t*-test at the 5% level of significance (*p* < 0.05), utilizing the statistical software package SPSS 18.0 (SPSS, Chicago, IL, USA). The concordance between the mathematical models and the experimental data was evaluated using the adjusted coefficient of determination (*R*^2^) and Akaike Information Criterion (AIC) value.

## 3. Results

### 3.1. Photosynthesis and Stomatal Conductance Response Under Variable Irradiance

The light-response curves of photosynthesis (*A*_n_–*I*) for the three species in question followed the anticipated pattern. At high irradiance levels, *T. repens* showed a slight decline in *A*_n_ beyond 1 200 μmol·m^−2^·s^−1^, suggesting the occurrence of dynamic down-regulation or photoinhibition (Figure 1A). In the case of *L. perenne* and *T. aestivum*, the *A*_n_ tends to reach saturation as light intensity increases under high light intensity conditions (Figure 1B,C).

In the present study, Equations (2), (5) and (7) are capable of describing the tendency of the stomatal conductance-response curves of photosynthesis (*A*_n_–*g*_sc_) for the three species at a CO_2_ concentration of 420 μmol·mol^−1^ (Figure 2). It can be clearly observed that *A*_n_ increases with rising *g*_sc_, yet the relationship between *A*_n_ and *g*_sc_ is nonlinear, particularly at high *A*_n_ values. Upon examination of Figure 2, it is apparent that the fitted curves from Equations (2) and (5) diverge from the observed data points, with the most significant deviations occurring at high *A*_n_ values. In contrast, Equation (7) provides a robust characterization of the *A*_n_–*g*_sc_ relationship for *L. perenne* and *T. aestivum*, yielding extremely high determination coefficients (*R*^2^) and low AIC values (Table 1), indicating a strong fit to the observed data points.

The comparative analysis of the *g*_0_ and *g*_1_ parameter values estimated by Equations (2), (5) and (7) shows that for the same species, there is no significant difference in *g*_0_ values derived from Equations (2) and (5) (*p* > 0.05), while the *g*_0_ value derived from Equation (7) is significantly lower than those obtained from Equations (2) and (5) (*p* < 0.05, Table 1). Given that the dimension of *g*_1_ varies between Equations (2), (5) and (7), Equation (2) (BWB model) provides the numerically highest *g*_1_ values for the three species. Furthermore, the fitting values derived from Equation (7) indicate that the most accurate fit is achieved for *L. perenne* and *T. aestivum*.

### 3.2. Photosynthesis and Stomatal Conductance Response Under Variable CO_2_ Concentration at I_sat_

The CO_2_-response curves of photosynthesis (*A*_n_–*C*_a_) for the three species under investigation adhered to the expected trends (Figure 3). *T. repens* demonstrated a marked increase in *A*_n_ in response to elevated CO_2_ levels (Figure 3A). For *L. perenne* and *T. aestivum*, the *A*_n_ approached saturation as CO_2_ concentration increased under high CO_2_ conditions, signifying a plateau in photosynthetic capacity and the onset of triose phosphate utilization (TPU) limitation (Figure 3B,C).

Under saturating irradiance conditions, the relationship between *A*_n_ and *g*_sc_ for the three species is shown in Figure 4. It is clear that *A*_n_ decreases as *g*_sc_ increases except for *T. aestivum*, but the relationship between *A*_n_ and *g*_sc_ is nonlinear, especially at high *A*_n_ values. In addition, although Equations (2), (5) and (7) can all describe the tendency of the *A*_n_–*g*_sc_ curves for the three species under conditions of their respective saturating irradiance, the concordance between the mathematical models and the observed data shows obvious differences. A close look at Figure 4 reveals that the fitted curves from Equations (2) and (5) deviate from the observed data points, with the most notable discrepancies at low *A*_n_ values for *T. repens* (Figure 4A), at both low and high *A*_n_ values for *L. perenne* (Figure 4B), and at high *A*_n_ values for *T. aestivum* (Figure 4C). In contrast, Equation (7) offers a reliable representation of the *A*_n_–*g*_sc_ relationship for all three species, resulting in extremely high *R*^2^ and low AIC values (Table 2), which signifies an excellent fit to the empirical data.

The comparative analysis of the *g*_0_ and *g*_1_ parameter values estimated by Equations (2), (5) and (7) shows that, there is no significant difference in *g*_0_ values derived from Equations (2), (5) and (7) for *L. perenne* and *T. aestivum* (*p* > 0.05), while the *g*_0_ value derived from Equation (7) is significantly higher than those obtained from Equations (2) and (5) for *T. repens* (*p* < 0.05, Table 2). Notably, the dimension of *g*_1_ differs between Equations (2), (5) and (7); hence, Equation (2) (BWB model) yields the highest numerical *g*_1_ values among the three species. Moreover, the fitting values derived from Equation (7) suggest that the optimal fit is obtained for all three species.

### 3.3. Photosynthesis and Stomatal Conductance Response Under Variable CO_2_ Concentration at Half of I_sat_

The *A*_n_–*C*_a_ relationship for the species under investigation followed the anticipated trends (Figure 5). *T. repens* exhibited a significant increase in *A*_n_ in response to elevated CO_2_ levels; however, *A*_n_ declined at a concentration of 1600 μmol·mol^−1^ (Figure 5A). In the case of *L. perenne* and *T. aestivum*, an approached saturation as the CO_2_ concentration increased under high CO_2_ conditions, indicating a plateau in photosynthetic capacity and the initiation of triose phosphate utilization (TPU) limitation (Figure 5B,C).

The relationship between *A*_n_ and *g*_sc_ for the three species under conditions of half their respective saturating irradiance can be observed in Figure 6. It is evident that the relationship between *A*_n_ and *g*_sc_ is complex and nonlinear. Although Equations (2), (5) and (7) can all describe the tendency of the *A*_n_–*g*_sc_ curves for the three species under conditions of half their respective saturating irradiance, the concordance between the mathematical models and the observed data shows obvious differences (Figure 6). Upon close examination of Figure 6, it becomes apparent that the fitted curves from Equations (2) and (5) diverge from the observed data points, particularly at low *A*_n_ values for *T. repens* (Figure 6A), and across a range of *A*_n_ values for *L. perenne* (Figure 6B) and *T. aestivum* (Figure 6C). Conversely, Equation (7) provides a reliable representation of the *A*_n_–*g*_sc_ relationship for all three species, yielding extremely high *R*^2^ values and low AIC values (Table 3), which indicate an excellent fit to the observed data for *L. perenne* (Figure 6B) and *T. aestivum* (Figure 6C).

The comparative analysis of the *g*_0_ and *g*_1_ parameter values estimated by Equations (2), (5) and (7) shows that, no significant difference is observed in *g*_0_ values derived from Equations (2), (5) and (7) for all three species under conditions of half their respective saturating irradiance (*p* > 0.05, Table 3). Notably, the dimension of *g*_1_ differs between Equations (2), (5) and (7); hence, Equation (2) (BWB model) yields the highest numerical *g*_1_ values among the three species. Moreover, the fitting values derived from Equation (7) suggest that the optimal fit is obtained for *L. perenne* and *T. aestivum*.

## 4. Discussions

The present study offers a detailed analysis of stomatal conductance (*g*_sc_) in three species—*T. repens*, *L. perenne*, and *T. aestivum*—across a range of environmental conditions. Our findings underscore the intricacy of *g*_sc_ regulation and its response to variations in light intensity and CO_2_ concentration, aligning with the broader understanding of stomatal behavior as a key integrator of plant responses to the environmental factors [2,37].

### 4.1. Theoretical Framework and Innovation of the New Model by Ye et al. [2]

The new model proposed by Ye et al. (Equation (7)) inherits its mathematical form from Fick’s first law, which ensures a solid physical foundation [33,34,35]. However, its core innovation lies in extending this physical law—which describes ideal gas diffusion—into a parameterized modeling framework capable of capturing complex plant physiological regulation. The key distinction resides in the introduction and interpretation of the parameters *g*_0_ and *g*_1_. In Fick’s law, the proportional relationship between *A*_n_ and *g*_sc_ (*A*_n_–*g*_sc_) is fixed at one, assuming stomata are the sole limiting factor for photosynthesis and that the plant operates in a theoretically optimal state [2,33]. However, real-world plant physiological processes often cause systematic deviations in photosynthetic behavior. By introducing the fitting parameters *g*_1_ and *g*_0_, Equation (7) effectively captures these deviations (Figure 2, Figure 4 and Figure 6). In contrast, the BWB model, being highly empirical, lacks a clear physiological interpretation for its parameter *g*_1_, while the Medlyn model, grounded in optimization theory, shows limitations in dynamic responses and structural acclimation—as evidenced in this study by its clear deviations under high light or low CO_2_ conditions (Figure 2, Figure 4 and Figure 6). Therefore, the novelty of the model proposed by Ye et al. is not merely an algebraic rearrangement of Fick’s first law; rather, it enables us to effectively quantify the regulatory effects of non-stomatal limitations—such as *g*_m_ or biochemical processes—on the photosynthesis-stomatal conductance relationship within a unified framework.

### 4.2. Model Comparison and Fit

The comparative analysis of the BWB, Medlyn, and the new models has revealed significant discrepancies in their ability to capture the relationship between the net photosynthetic rate (*A*_n_) and stomatal conductance to CO_2_ (*g*_sc_). The BWB model, a cornerstone in the field, yielded the highest *g*_1_ values, suggesting an overly steep slope in the *g*_sc_ response. This characteristic may not accurately encapsulate the nuanced physiological mechanisms that govern stomatal behavior, as recent research has indicated [5,11,27,38]. The curves generated by the Medlyn et al. [26] model diverged from the observed data points at high *A*_n_ for the three species, suggesting that the model may not fully capture the dynamic stomatal responses under diverse environmental conditions.

In contrast, Equation (7) emerged as a more robust descriptor of the *A*_n_–*g*_sc_ relationship, delivering high *R*^2^ values and low AIC values across all plant species under variable irradiance and CO_2_ concentrations. This outcome indicates a superior alignment with empirical data, underscoring the equation’s potential as a more accurate predictive tool. This aligns with the call for models that better represent stomatal dynamics and account for the complex interactions between environmental stressors and plant physiology.

Equation (7)’s success in capturing the *A*_n_–*g*_sc_ relationship may be attributed to its ability to incorporate a broader range of environmental variables and plant physiological responses, providing a more comprehensive framework for understanding stomatal behavior. This is particularly important in the context of global change, where plants are subjected to increasing atmospheric CO_2_ concentrations, temperature variability, and drought stress. The model’s robustness across different plant species suggests its versatility and potential applicability in diverse ecosystems and under varying environmental conditions.

However, it is important to note that while Equation (7) shows promise, there is still a need for further research to refine and validate these models under a wider range of conditions. This includes understanding how plants acclimate structurally and physiologically to persistent changes in atmospheric CO_2_, and how these acclimations interact with other environmental variables such as light and temperature [23]. Additionally, this model must be tested against a broader dataset, including different plant species and environmental scenarios, to ensure their predictive accuracy and applicability in actual growth environment conditions.

### 4.3. Photosynthetic Response and Stomatal Conductance

The photosynthetic response to fluctuating irradiance and CO_2_ concentrations was consistent with anticipated patterns, particularly for *T. repens*, which showed a significant enhancement in net photosynthetic rate (*A*_n_) in response to increased CO_2_ levels (Figure 1, Figure 3 and Figure 5). Notably, at a CO_2_ concentration of 420 μmol·mol^−1^, *T. repens* exhibited a decline in *A*_n_ at a light intensity of 1200 μmol·m^−2^·s^−1^ (Figure 1A), suggesting the potential engagement of photoinhibition or dynamic down-regulation mechanisms [30]. These responses may act as protective measures against excessive irradiance exposure, as previously detailed by Murchie and Niyogi [39]. In the case of *L. perenne* and *T. aestivum*, the *A*_n_ under high irradiance levels indicated a plateau in photosynthetic capacity, signifying the occurrence of photosynthetic saturation (Figure 1B,C). Moreover, both species displayed a saturation trend in *A*_n_ under high CO_2_ conditions at their respective saturating irradiance (*I*_sat_) and half of *I*_sat_ (Figure 2B,C and Figure 3B,C), indicating a plateau in photosynthetic capacity and the initiation of triose phosphate utilization (TPU) limitation. This TPU limitation is a pivotal factor influencing the photosynthetic performance of plants in environments with elevated CO_2_ levels [12,26,29,34,40].

Furthermore, this study numerically disclosed substantial variability in the *g*_0_ and *g*_1_ parameters as estimated by Equations (2), (5) and (7) among the three species in question. Notably, Equation (2) yielded *g*_0_ values that extended from −0.280 to 0.087 mol·m^−2^·s^−1^, with corresponding *g*_1_ values spanning a broad range of 3.446 to 34.391 (Table 1, Table 2 and Table 3). When applying Equation (5), the *g*_0_ values were found to range between −0.586 to 0.085 mol·m^−2^·s^−1^, and *g*_1_ values fluctuated between 0.209 and 21.427 (Table 1, Table 2 and Table 3). Equation (7) provided *g*_0_ estimates that fell within a narrower band, from −0.264 to 0.000 mol·m^−2^·s^−1^, and *g*_1_ values that were comparatively stable, extending from 0.724 to 1.952 (Table 1, Table 2 and Table 3). Strikingly, Equation (7), with the exception of *T. aestivum* at *I*_sat_, produced *g*_0_ values tightly clustered between −0.013 and 0.000 mol·m^−2^·s^−1^, and *g*_1_ values that were remarkably consistent, centering around 0.8 (Table 1, Table 2 and Table 3). The remarkable stability of *g*_1_ strongly suggests that it may represent a conserved parameter characterizing the intrinsic coupling of the stomatal-photosynthetic system in C_3_ plants. When *g*_1_ ≈ 0.8, it implies that under measured conditions, the net photosynthetic rate is approximately 80% of the value predicted by the ideal Fick’s law of diffusion. This systematic “discount” can reasonably be attributed to the combined effects of limitations in *g*_m_ and other biochemical processes. Therefore, *g*_1_ can be interpreted as a “comprehensive physiological regulation factor,” whose magnitude directly reflects the degree to which internal leaf physiological resistances attenuate ideal gas exchange. This interpretation endows *g*_1_ with clear biological significance, surpassing the purely empirical nature of the parameter *g*_1_ in the BWB model while also circumventing the dimensional issues associated with *g*_1_ in the Medlyn model. This consistency points to a reliable and predictable stomatal response across the species under scrutiny, particularly when subjected to varying irradiance levels and CO_2_ concentrations. These observations highlight the diversity in stomatal conductance parameters and emphasize the importance of model-specific analysis when interpreting plant responses to environmental scenarios.

Conversely, *g*_0_ is recognized as the residual stomatal conductance in several models [2,18,22,26]. Typically, *g*_0_ is considered to be minuscule and is often assumed to be zero in many models [2,28,29]. However, our study reveals that for both the BWB and Medlyn models, *g*_0_ exceeds zero significantly for the three plant species under varying light irradiance and CO_2_ concentrations, as detailed in Table 1, Table 2 and Table 3, indicating that its impact cannot be overlooked. In the context of Equation (7), *g*_0_ is relatively smaller and can generally be considered negligible, except for *T. aestivum* under saturating light intensities (Table 1, Table 2 and Table 3). Consequently, our findings suggest that it may be more accurate to view *g*_0_ as a fitting constant rather than residual stomatal conductance. Otherwise, it would be difficult to explain why the estimated values of *g*_0_ for *T. aestivum* under saturating light conditions, as calculated by Equations (2) and (7), are below −0.2 mol·m^−2^·s^−1^ (Table 2), and for *T. aestivum* under half-saturating light conditions, as calculated by Equations (2) and (5), are also below −0.2 mol·m^−2^·s^−1^ (Table 3). Regarding the occurrence of negative *g*_0_ values obtained from model fittings, particularly the frequent instances observed with the BWB and Medlyn model fits (Table 1, Table 2 and Table 3), these are likely mathematical artifacts arising from the fitting processes of the respective formulas. The functional forms of these models may force the fitting procedure to produce a negative intercept to achieve the best possible fit to the nonlinear data, thereby resulting in *g*_0_ values that contradict biological reality. Consequently, the *g*_0_ parameters derived from these model fits may not reflect the true physiological reality of plant stomata, as stomatal conductance cannot be negative [2,32]. Such negative *g*_0_ values may occur when the model attempts to minimize residuals under conditions where stomatal conductance is very low or approaching zero [18,26]. Furthermore, we note that compared to Equation (7), lower negative *g*_0_ values are more commonly obtained when fitting the *A*_n_–*C*_a_ response curves for the three C_3_ species under half-saturating irradiance using Equations (2) and (5). This observation further supports the need for a more physiologically consistent model.

### 4.4. Limitations and Future Perspectives

While this study presents a promising model for stomatal conductance, it is important to acknowledge its limitations, which also outline clear pathways for future research. Firstly, the evaluation and parameterization of the models are conducted on three C_3_ species (*T. repens*, *L. perenne*, and *T. aestivum*). Although this provides a robust comparison within a specific plant species, the generalizability of our findings to other species, particularly woody plants, C_4_ species, or plants from extreme habitats, remains to be tested. Future work should include a broader phylogenetic range of species to validate and potentially refine the model across diverse plant kingdoms. Secondly, our measurements were performed under controlled, steady-state conditions in a gas exchange chamber. While this allows for precise isolation of the effects of irradiance and CO_2_, it does not fully capture the dynamic and fluctuating nature of the natural environment, such as rapid changes in light (e.g., sunflecks), wind, and vapor pressure deficit. Testing the model’s performance under field conditions with real-world environmental variability is a crucial next step. Finally, the model was parameterized and validated using the same dataset. To unequivocally establish its predictive power and avoid overfitting, an independent validation with a completely separate dataset is essential. Future research should prioritize applying the model to an independent dataset collected from different growth conditions or by independent research groups. Addressing these limitations will be critical for advancing the model from a promising theoretical framework to a robust tool for predicting plant-water-carbon interactions in real-world ecosystems under global change.

## 5. Conclusions

This study systematically evaluated the predictive performance of three stomatal conductance models—BWB, Medlyn, and a new model proposed by Ye et al. (Equation (7))—under variable light intensity and CO_2_ conditions for three C_3_ plant species. The results indicated that the traditional BWB and Medlyn models exhibited significant deviations in describing the relationship between *A*_n_ and *g*_sc_, particularly under conditions of high *A*_n_ or extreme environments. In contrast, the new model demonstrated exceptional robustness and accuracy, achieving high *R*^2^ values and low AIC values between simulated and measured data across all three species and conditions. Furthermore, the analysis of key parameters revealed that the fitted parameters (*g*_0_ and *g*_1_) in the new model showed superior consistency and stability. The *g*_0_ values obtained from the new model were notably more stable, supporting their interpretation as a fitting constant rather than a fixed physiological parameter. This characteristic stands in clear contrast to the often non-negligible or negative *g*_0_ estimates from the BWB and Medlyn models, clearly challenging the conventional concept of “residual conductance”. Therefore, the new model provides a more reliable tool for simulating stomatal behavior. It offers significant value for improving predictions of crop water use efficiency and understanding plant responses to global change. However, while the new model performs well across the three C_3_ species tested, further validation across a wider range of species and environmental conditions is necessary before broader generalizations can be made.

## Figures and Tables

**Figure 1 biology-14-01501-f001:**
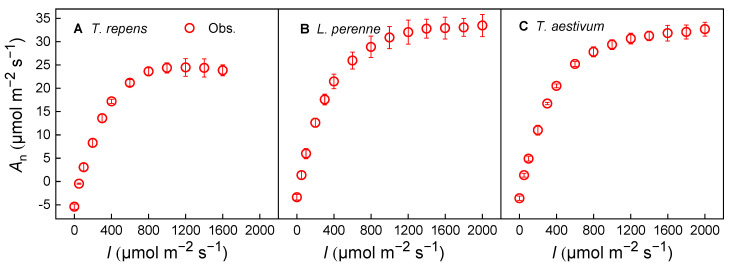
Light-response curves of photosynthesis (*A*_n_–*I*) for *T. repens* (**A**), *L. perenne* (**B**) and *T. aestivum* (**C**) at a CO_2_ concentration of 420 μmol·mol^−1^. *A*_n_: net photosynthetic rate; *I*: light intensity; *T. repens*: *Trifolium repens*; *L. perenne*: *Lolium perenne*; *T. aestivum*: *Triticum aestivum*. The open symbols on the graph denote the measured data points. The data are depicted as the mean ± standard error (*SE*), based on three replicates (*n* = 3).

**Figure 2 biology-14-01501-f002:**
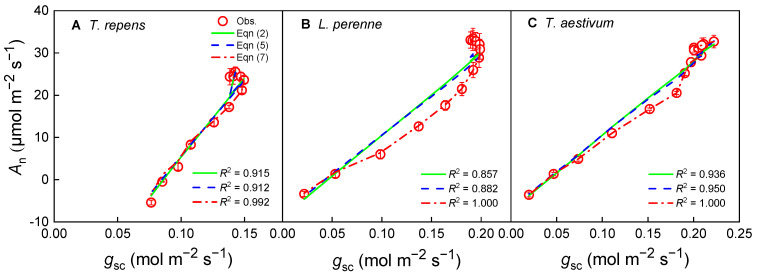
Stomatal conductance-response curves of photosynthesis (*A*_n_–*g*_sc_) for *T. repens* (**A**), *L. perenne* (**B**) and *T. aestivum* (**C**) at a CO_2_ concentration of 420 μmol·mol^−1^. *A*_n_: net photosynthetic rate; *g*_sc_: stomatal conductance to CO_2_; *T. repens*: *Trifolium repens*; *L. perenne*: *Lolium perenne*; *T. aestivum*: *Triticum aestivum*. The open symbols represent the measured data points, while the solid line indicates the model fit derived from Equation (2). The dashed line corresponds to the fit based on Equation (5), and the dash-dot line signifies the fit according to Equation (7). The data points are expressed as the mean ± standard error (*SE*), with *n* = 3 replicates.

**Figure 3 biology-14-01501-f003:**
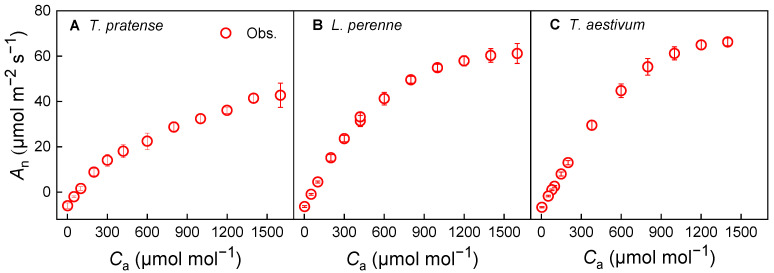
CO_2_-response curves of photosynthesis (*A*_n_–*C*_a_) for *T. repens* (**A**), *L. perenne* (**B**) and *T. aestivum* (**C**) at the saturating irradiance. *A*_n_: net photosynthetic rate; *C*_a_: atmospheric CO_2_ concentration; *T. repens*: *Trifolium repens*; *L. perenne*: *Lolium perenne*; *T. aestivum*: *Triticum aestivum*. The open symbols on the graph denote the measured data points. The data are depicted as the mean ± standard error (*SE*), based on three replicates (*n* = 3).

**Figure 4 biology-14-01501-f004:**
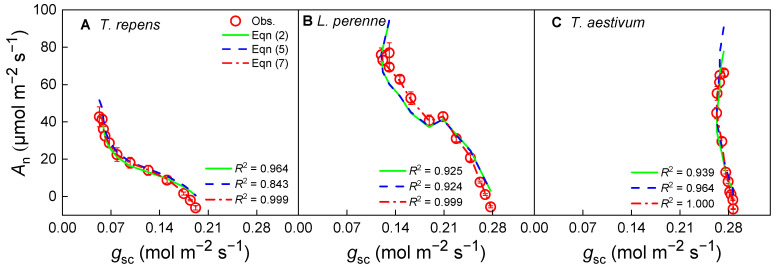
Stomatal conductance-response curves of photosynthesis (*A*_n_–*g*_sc_) for *T. repens* (**A**), *L. perenne* (**B**) and *T. aestivum* (**C**) at the saturating irradiance. *A*_n_: net photosynthetic rate; *g*_sc_: stomatal conductance to CO_2_; *T. repens*: *Trifolium repens*; *L. perenne*: *Lolium perenne*; *T. aestivum*: *Triticum aestivum*. The open symbols represent the measured data points, while the solid line indicates the model fit derived from Equation (2). The dashed line corresponds to the fit based on Equation (5), and the dash-dot line signifies the fit according to Equation (7). The data points are expressed as the mean ± standard error (*SE*), with *n* = 3 replicates.

**Figure 5 biology-14-01501-f005:**
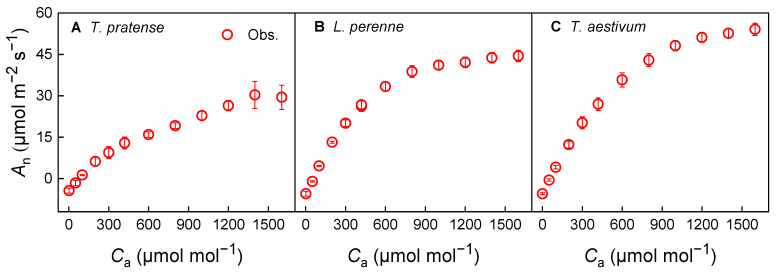
CO_2_-response curves of photosynthesis (*A*_n_–*C*_a_) for *T. repens* (**A**), *L. perenne* (**B**) and *T. aestivum* (**C**) at half of their corresponding saturating irradiance. *A*_n_: net photosynthetic rate; *C*_a_: atmospheric CO_2_ concentration; *T. repens*: *Trifolium repens*; *L. perenne*: *Lolium perenne*; *T. aestivum*: *Triticum aestivum*. The open symbols on the graph denote the measured data points. The data are depicted as the mean ± standard error (*SE*), based on three replicates (*n* = 3).

**Figure 6 biology-14-01501-f006:**
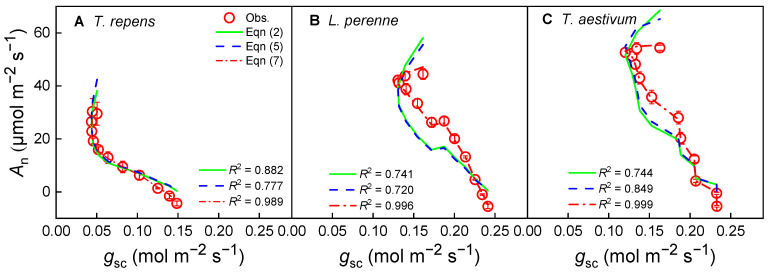
Stomatal conductance-response curves of photosynthesis (*A*_n_–*g*_sc_) for *T. repens* (**A**), *L. perenne* (**B**) and *T. aestivum* (**C**) at half of their corresponding saturating irradiance. *A*_n_: net photosynthetic rate; *g*_sc_: stomatal conductance to CO_2_; *T. repens*: *Trifolium repens*; *L. perenne*: *Lolium perenne*; *T. aestivum*: *Triticum aestivum*. The open symbols represent the measured data points, while the solid line indicates the model fit derived from Equation (2). The dashed line corresponds to the fit based on Equation (5), and the dash-dot line signifies the fit according to Equation (7). The data points are expressed as the mean ± standard error (*SE*), with *n* = 3 replicates.

**Table 1 biology-14-01501-t001:** The estimates by Equations (2), (5) and (7) for three C_3_ species when light-response curves of photosynthesis were performed under a CO_2_ concentration of 420 μmol·mol^−1^ (mean ± *SE*, *n* = 3), respectively. Estimated values within one plant which are statistically significantly different (*p* < 0.05) are annotated with different superscript letters, with the same superscript letter indicating no difference for Equations (2), (5) and (7). The unit of *g*_0_ is mol·m^−2^·s^−1^.

	*T. repens*			*L. perenne*			*T. aestivum*		
	Equation (2)	Equation (5)	Equation (7)	Equation (2)	Equation (5)	Equation (7)	Equation (2)	Equation (5)	Equation (7)
*g* _0_	0.087 ± 0.008 ^a^	0.085 ± 0.008 ^a^	−0.002 ± 0.005 ^b^	0.045 ± 0.009 ^a^	0.042 ± 0.009 ^a^	−0.002 ± 0.000 ^b^	0.040 ± 0.012 ^a^	0.037 ± 0.010 ^a^	−0.002 ± 0.000 ^b^
*g* _1_	3.446 ± 0.294	0.209 ± 0.209	0.756 ± 0.039	4.515 ± 0.065	1.499 ± 0.089	0.727 ± 0.004	4.574 ± 0.455	1.513 ± 0.223	0.724 ± 0.006
*R* ^2^	0.914	0.912	0.992	0.857	0.882	1.000	0.936	0.950	1.000
AIC	12.08	12.07	10.18	21.01	18.57	4.32	26.75	25.53	3.91

**Table 2 biology-14-01501-t002:** The estimates by Equations (2), (5) and (7) for three C_3_ species when CO_2_-response curves of photosynthesis were performed at their corresponding saturating irradiance (mean ± *SE*, *n* = 3), respectively. Estimated values within one plant which are statistically significantly different (*p* < 0.05) are annotated with different superscript letters, with the same superscript letter indicating no difference. The unit of *g*_0_ is mol·m^−2^·s^−1^.

	*T. repens*			*L. perenne*			*T. aestivum*		
	Equation (2)	Equation (5)	Equation (7)	Equation (2)	Equation (5)	Equation (7)	Equation (2)	Equation (5)	Equation (7)
*g* _0_	−0.162 ± 0.051 ^a^	−0.105 ± 0.005 ^a^	−0.002 ± 0.000 ^b^	0.000 ± 0.039 ^a^	0.018 ± 0.028 ^a^	−0.001 ± 0.003 ^a^	−0.280 ± 0.220 ^a^	0.035 ± 0.079 ^a^	−0.264 ± 0.157 ^a^
*g* _1_	34.391 ± 8.301	6.701 ± 1.599	0.742 ± 0.015	8.633 ± 1.613	2.214 ± 0.564	0.740 ± 0.018	17.406 ± 7.652	4.572 ± 2.820	1.952 ± 0.695
*R* ^2^	0.964	0.843	0.999	0.925	0.924	0.999	0.939	0.964	1.000
AIC	22.39	19.21	6.06	26.27	22.87	9.36	23.68	23.42	8.69

**Table 3 biology-14-01501-t003:** The estimates by Equations (2), (5) and (7) for the three C_3_ species when CO_2_-response curves of photosynthesis were performed at half of their corresponding saturating irradiance (mean ± *SE*, *n* = 3), respectively. Within a single plant species, estimated values that are statistically distinct (*p* < 0.05) are marked with unique superscript letters, while identical superscript letters signify no significant difference. The unit of *g*_0_ is mol·m^−2^·s^−1^.

	*T. repens*			*L. perenne*			*T. aestivum*		
	Equation (2)	Equation (5)	Equation (7)	Equation (2)	Equation (5)	Equation (7)	Equation (2)	Equation (5)	Equation (7)
*g* _0_	−0.082 ± 0.038 ^a^	−0.033 ± 0.015 ^a^	−0.002 ± 0.000 ^a^	−0.324 ± 0.176 ^a^	−0.312 ± 0.175 ^a^	−0.005 ± 0.003 ^a^	−0.269 ± 0.150 ^a^	−0.586 ± 0.309 ^a^	−0.013 ± 0.008 ^a^
*g* _1_	25.502 ± 5.474	4.853 ± 1.891	0.760 ± 0.046	41.855 ± 15.809	18.901 ± 8.429	0.782 ± 0.025	22.981 ± 10.228	21.427 ± 10.123	0.811 ± 0.067
*R* ^2^	0.882	0.777	0.989	0.741	0.720	0.996	0.744	0.849	0.999
AIC	21.98	19.61	12.18	25.74	24.79	12.02	27.94	27.81	10.05

## Data Availability

The raw data supporting the conclusions of this article will be made available by the authors on request.

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
