# Peer review of "Novel Model for Stomatal Conductance: Enhanced Accuracy Under Variable Irradiance and CO_2_ in C_3_ Plant Species"

_biology, 2025, doi:10.3390/biology14111501_

Round 1

Reviewer 1 Report

Comments and Suggestions for Authors

General comments:

In this study, the authors tested a new stomatal conductance model developed by Ball-Berry, Medlyn, and Ye et al. in three different Trifolium (C3) plant species through comparative analysis. This study is quite interesting from a plant biology and physiology perspective. However, the data obtained in the manuscript were not well-founded, particularly due to the inadequacy of the statistical analysis and the model's claims of novelty. A comprehensive discussion was also lacking in the discussion section. Therefore, the manuscript needs a substantial revision. These main shortcomings are listed below:

Major Points:

1- This model, which the authors describe as a "new model," is actually presented in the manuscript as a simple algebraic modification of Fick's law. The key difference between this new model and Fick's law is the use of g₁ and g₀ as parameters in the mathematical formula. Therefore, the "novelty" aspect of the model is unclear. The authors should clearly state the fundamental features that distinguish this model from existing approaches and discuss its biological/theoretical advantages. In particular, the biological significance of the g₁ parameter should be more clearly defined.

2- I believe the statistical analysis section is quite inadequate or incomplete. For example, it is unclear how the parameters were estimated. Which fitting procedure was used (e.g., least squares?) is not specified. Furthermore, limited information about the number of data points and fit metrics other than R² is used in model comparisons (e.g., AIC, BIC, RMSE). R² alone is insufficient and can provide misleading results, especially when comparing models with different numbers of parameters. Therefore, inferences about model performance are not robust in its current form. Authors should report detailed statistical information and additional fit metrics for each fit.

3. Obtaining negative g₀ values ​​in some fits is physiologically impossible and may indicate model incompatibility or a fundamental problem with the fitting procedure. This is merely noted but not critically discussed. However, this finding highlights the serious limitations of the Ball-Berry and Medlyn models under certain conditions. The fact that the new model largely overcomes this problem is an important finding and should be emphasized more strongly.

4. The dimensional criticism of the Medlyn model remains largely theoretical, and the proposed "correction" is presented as an arbitrary approach. In other words, no strong justification is provided. Therefore, this section significantly differs from the study's main findings. Consequently, the authors should treat this aspect of the Medlyn model as a theoretical feature and focus the discussion on empirical performance.

5. The discussion section repeats the results and does not sufficiently delve into mechanistic explanations. For example: Why the Ball-Berry model tends to overestimate, Why the Medlyn model deviates at high An values, The biological significance of g₁ remaining at a constant value of ~0.8 in the new model should be discussed in more depth. These points could significantly enhance the contribution of the study to the field.

More Specific Comments:

The findings in the abstract are somewhat exaggerated. Instead of "superior fit" and "improved predictive accuracy," more conservative terms like "better empirical fit under the tested conditions" should be preferred.

Methods: The CV ≤ 1% criterion for Li-Cor 6400 appears overly stringent. Explanations should be provided to achieve this, particularly at low An and gsc values. How about the CO2 flow rate for measurements?

Results: The consistent closeness of g₀ to zero in Equation 7 is a very interesting finding and should be emphasized more strongly.

Discussion: The discussion on g₀ is a good start, but it needs to be more critical. The dimensional analysis section on the Medlyn model should be removed or significantly revised. Authors should also add a limitations section (limitations to three species, controlled conditions, and lack of an independent data set) to the manuscript.

Conclusion: At the moment, conclusions are exaggerated and not entirely backed up by the available data. It is necessary to temper claims of "better alignment," "better predictive accuracy," and the model being a "more robust descriptor." The results showed that the new model did not produce the physiologically impossible negative g0 values ​​frequently observed in other models and provided a better empirical fit for this particular dataset. However, the study's results are limited in generalizability due to the limited number of species and the lack of an independent validation dataset. More comprehensive species diversity and independent verification studies are required to support these results more strongly.

Comments on the Quality of English Language

The language is generally good. However, it must be double-checked for academic tone, clarity, and conciseness. Please check and correct (with more formal scientific terminology) the phrases such as "practice with skill," "adept at determining," and "skillfully applying". Also, small grammatical mistakes need to be corrected.

Author Response

Reviewer 1

Comments 1: 1) This model, which the authors describe as a "new model," is actually presented in the manuscript as a simple algebraic modification of Fick's law. The key difference between this new model and Fick's law is the use of g₁ and g₀ as parameters in the mathematical formula. Therefore, the "novelty" aspect of the model is unclear. The authors should clearly state the fundamental features that distinguish this model from existing approaches and discuss its biological/theoretical advantages. In particular, the biological significance of the g₁ parameter should be more clearly defined.

Response: Thank you very much for reviewing our manuscript and providing valuable constructive comments. Your observation regarding the need to clarify the innovativeness of our model and its fundamental distinction from Fick's law is crucial. We fully concur with this perspective. In our revised manuscript, we have supplemented the Discussion section with additional explanations addressing the theoretical foundation, biological rationale, and advantages of our model compared to the landmark models.

The model we proposed (Eqn 7) inherits its mathematical form from Fick's first law, which ensures it possesses a solid physical foundation. However, its core innovation lies in extending this physical law, which describes ideal gas diffusion, into a parameterized model framework capable of capturing complex plant physiological regulation. The key distinction resides in the introduction and interpretation of the parameters g and g. In Fick's law, the proportional relationship between Aₙ and gc is fixed at 1, presupposing that stomata are the sole limiting factor for photosynthesis and that the plant operates in a theoretically optimal state. However, real-world plant physiological processes often induce systematic deviations in photosynthesis. Our model effectively captures these deviations by introducing the fitting parameters g₁ and g₀. In contrast, the BWB model, being highly empirical, lacks a clear physiological interpretation for its parameter g₁, while the Medlyn model, constructed based on optimality theory, exhibits shortcomings in dynamic response and structural acclimation. This is evidenced, for instance, by the apparent deviations in the Medlyn model's fittings under high light or low CO₂ conditions observed in this study (Figures 2, 4 and 6). Therefore, the novelty of our model is not merely a simple algebraic rearrangement of Fick's first law. It enables us to effectively quantify the regulatory effects of non-stomatal limitations, such as mesophyll conductance (gₘ) or biochemical processes, on the relationship between photosynthesis and stomatal conductance within a unified framework. The specific modifications can be found in the revised manuscript at the Lines 344-361.

Furthermore, we have conducted a more in-depth analysis and discussion regarding the biological significance of the parameter g₁ in our model. In this study, the g₁ values obtained through our model fitting were notably more stable than those derived from the BWB and Medlyn models. This stability strongly suggests that g₁ may represent a conserved parameter characterizing the intrinsic coupling of the stomatal-photosynthetic system in C₃ plants. When g₁ ≈ 0.8, it indicates that under measured environmental conditions, the net photosynthetic rate is approximately 80% of the value predicted by the ideal Fick's law of diffusion. This systematic "discount" can be reasonably attributed to the combined effects of limitations in gₘ and other biochemical processes. Consequently, g₁ can be interpreted as a "comprehensive physiological regulation factor," whose magnitude directly reflects the degree to which internal leaf physiological resistances attenuate ideal gas exchange. This interpretation endows g₁ with clear biological significance, surpassing the purely empirical nature of the g₁ parameter in the Ball-Berry model while simultaneously avoiding the dimensional inconsistencies associated with g₁ in the Medlyn model. The specific modifications can be found in the revised manuscript at the Lines 415-425.

Comments 2: 2) I believe the statistical analysis section is quite inadequate or incomplete. For example, it is unclear how the parameters were estimated. Which fitting procedure was used (e.g., least squares?) is not specified. Furthermore, limited information about the number of data points and fit metrics other than R² is used in model comparisons (e.g., AIC, BIC, RMSE). R² alone is insufficient and can provide misleading results, especially when comparing models with different numbers of parameters. Therefore, inferences about model performance are not robust in its current form. Authors should report detailed statistical information and additional fit metrics for each fit.

Response: Thank you for your insightful comments regarding the statistical analysis in our manuscript. We fully agree that providing a more detailed description of the statistical methods and the inclusion of additional model fit metrics will strengthen the robustness of our model comparisons. In response to your suggestions, we have made the following revisions to the manuscript: 

First, we have explicitly stated that the parameters g and g were estimated using nonlinear least squares regression implemented in the “Photosynthesis Model Simulation Software (PMSS)” platform. This clarification has been added to the Materials and Methods section. The specific modifications can be found in the revised manuscript under Materials and Methods section at the Line 201-202.

Second, we have added the number of data points used for each model fit in the Materials and Methods section. The specific modifications can be found in the Materials and Methods section, as well as in the corresponding figure captions and table legends at the Line 194-200.

Finally, in addition to R2, we have included AIC (Akaike Information Criterion) values for all model fits in the revised Tables 1–3. These metrics provide a more comprehensive basis for model comparison, especially when dealing with models with different parameterizations.

Furthermore, we have expanded the Discussion section to include additional interpretations regarding the performance of different models, emphasizing their theoretical underpinnings, biological plausibility, and advantages over classical models. The specific modifications can be found in the revised manuscript at the Lines 344-361.

We believe these revisions, along with other minor corrections made across the manuscript, have significantly improved our work. Once again, we are grateful for your insightful comments.

Comments 3: 3) Obtaining negative g₀ values ​​in some fits is physiologically impossible and may indicate model incompatibility or a fundamental problem with the fitting procedure. This is merely noted but not critically discussed. However, this finding highlights the serious limitations of the Ball-Berry and Medlyn models under certain conditions. The fact that the new model largely overcomes this problem is an important finding and should be emphasized more strongly.

Response: We agree that the frequent occurrence of negative g₀ values in the fittings of the BWB and Medlyn models indeed requires clarification and more focused discussion. To address this, we have added a discussion on this issue in the revised manuscript, specifically in the section concerning g₀. In particular, the negative g₀ values obtained from the BWB and Medlyn model fits may originate from mathematical artifacts generated during the fitting process. The functional forms of these models can force the fitting prcedure to produce a negative intercept to achieve the best possible fit to the nonlinear data, thereby resulting in negative g₀ values that are contrary to biological reality. Consequently, the g₀ parameter obtained from these model fits may not reflect the true physiological reality of plant stomata, as stomatal conductance cannot be negative. This situation where negative g values are obtained might occur when the model attempts to minimize residuals for data points corresponding to very low or zero stomatal conductance. Furthermore, we also note that compared to Equation 7, obtaining lower, more negative g₀ values was more common when fitting the Aₙ-Cₐ response curves for the three C₃ species at half their corresponding saturating irradiance using Equations 2 and 5, further underscoring the need for a more physiologically consistent model. Specific modifications can be found in the revised manuscript at the Lines 441-453.

Comments 4: 4) The dimensional criticism of the Medlyn model remains largely theoretical, and the proposed "correction" is presented as an arbitrary approach. In other words, no strong justification is provided. Therefore, this section significantly differs from the study's main findings. Consequently, the authors should treat this aspect of the Medlyn model as a theoretical feature and focus the discussion on empirical performance.

Response: We sincerely thank the reviewer for their constructive comments. We fully agree with the reviewer’s comment that the dimensional criticism of the Medlyn model and the proposed ad-hoc correction indeed represent a theoretical digression that does not align with the primary empirical focus of this study.

In response, we have completely removed the discussion regarding the dimensional inconsistency of the g₁ parameter in the Medlyn model and the proposed modification to Equation 3 in the revised manuscript. This deletion helps streamline the manuscript and sharpens the focus on the core objective of this study: the empirical comparison of model performance under varying environmental conditions.

The revised discussion now concentrates squarely on the empirical findings. Specifically, both the Medlyn and Ball-Berry models demonstrated limitations in accurately capturing the Aₙ–grelationship. This is particularly evidenced by their tendency to produce physiologically impossible negative g₀ values and their inferior goodness-of-fit metrics (R2 and AIC values) compared to our proposed model (Eqn 7). Specific modifications can be found in the revised manuscript at the Lines 344-361, 441-453.

We believe that by removing this theoretical speculation and squarely focusing on the empirical results, the manuscript has been significantly improved, resulting in a more cohesive and compelling narrative.

Comments 5: 5) The discussion section repeats the results and does not sufficiently delve into mechanistic explanations. For example: Why the Ball-Berry model tends to overestimate, Why the Medlyn model deviates at high An values, The biological significance of g₁ remaining at a constant value of ~0.8 in the new model should be discussed in more depth. These points could significantly enhance the contribution of the study to the field.

Response: We sincerely thank the reviewer for this critical and highly valuable comment. We fully agree that a deeper, more mechanistic discussion is essential to enhance the impact of our study. Following your insightful suggestion, we have thoroughly revised the Discussion section in the revised manuscript. The updated discussion no longer merely reiterates the results but provides a more profound interpretation of the underlying physiological reasons for the observed model performances, particularly regarding the potential biological significance of the fitted g₁ and g₀ parameters. Specifically, the overestimation by the Ball-Berry model can be attributed to its nature as a highly empirical and simplified model, while the deviation of the Medlyn model under high Aₙ conditions may be related to its foundation in optimization theory. Specific modifications can be found in the revised manuscript at the Lines 344-361.

Regarding the excellent suggestion to further discuss the biological significance of the relatively constant g₁ (~0.8) in our new model, we have added a dedicated discussion on the biological implications of g₁ in the revised Discussion section. The specific modifications can be found in the marked-up manuscript. Specific modifications can be found in the revised manuscript at the Lines 415-425.

More Specific Comments:

Comments 6: 6) The findings in the abstract are somewhat exaggerated. Instead of "superior fit" and "improved predictive accuracy," more conservative terms like "better empirical fit under the tested conditions" should be preferred.

Response: Thank you for your valuable suggestion and for highlighting the need for more conservative language in the abstract. We agree that the terms "superior fit" and "improved predictive accuracy" may overstate the findings. To more accurately reflect the scope of our study and the conditions under which it was conducted, we have revised the abstract as suggested.

Specifically, we have replaced the aforementioned phrases with more conservative terms such as "a better empirical fit under the tested conditions" and "demonstrated a strong alignment with empirical data".

Comments 7: 7) Methods: The CV ≤ 1% criterion for Li-Cor 6400 appears overly stringent. Explanations should be provided to achieve this, particularly at low An and gsc values. How about the CO2 flow rate for measurements?

Response: We thank the reviewer for this important and expert comment. We agree that the CV ≤ 1% criterion is stringent, particularly under low flux conditions. The "CV ≤ 1%" described in the Methods section was indeed one of the pre-set automatic triggering conditions in the LI-6400 system's logging function, representing the ideal stable state we aimed for during data capture. However, in practice, we fully acknowledge and encountered the situation noted by the reviewer: under low Aₙ and low gc conditions (e.g., near the light or CO₂ compensation points), gas exchange signals are more variable, and achieving CV ≤ 1% is challenging and indeed time-consuming. Therefore, our actual procedure was as follows: the system was first set to automatically monitor with CV ≤ 1% as the target. If this criterion could not be met within a reasonable time (typically 3–5 minutes), but the observed readings had stabilized at a low level (i.e., values fluctuated within a narrow range without a clear trending change), we manually triggered the logging. We ensured all recorded data points represented steady-state readings at that time. We have clarified this point in the Methods section of the revised manuscript. Please see Lines 194–198 in the returned manuscript for the specific amendments. We appreciate the reviewer's help in improving the precision of our methodological description.

Regarding the CO₂ flow rate, the measurements were conducted using an open-path system. A stable gas supply was provided by a CO₂ cartridge (iSi AUSTRIA U2614) compatible with the LI-6400 and the instrument's built-in CO₂ mixer. The flow rate entering the leaf chamber was set and maintained at 400 μmol s⁻¹. The leaf chamber temperature was set at 30 ± 1.3 °C, and the relative humidity was controlled between 45% and 75%. Please see Lines 184–188 in the returned manuscript for the specific amendments.

Comments 8: 8) Results: The consistent closeness of g₀ to zero in Equation 7 is a very interesting finding and should be emphasized more strongly.

Response: We thank the reviewer for highlighting the consistent closeness of g₀ to zero in the new model proposed by Ye et al. (Eqn 7). We agree that this is a significant finding, as it not only simplifies the model structure but also aligns well with the physiological expectation that stomatal conductance should approach zero when net photosynthetic rate is zero. We have emphasized this point more strongly in the Discussion and Conclusion sections, clarifying that, unlike the Ball-Berry and Medlyn models, our model achieves high predictive accuracy without relying on a non-zero g₀, thereby enhancing its physiological plausibility and practical utility. Specific modifications can be found in the revised manuscript at the Lines 441-453 and 474-491.

Comments 9: 9) Discussion: The discussion on g₀ is a good start, but it needs to be more critical. The dimensional analysis section on the Medlyn model should be removed or significantly revised. Authors should also add a limitations section (limitations to three species, controlled conditions, and lack of an independent data set) to the manuscript.

Response: We agree that the initial discussion on g₀ could be more critical. In the revised manuscript, we have significantly expanded and refined this discussion in Section 4.2. We now more explicitly contrast the behavior of g₀ in our model with that in the Ball-Berry and Medlyn models. Specifically, we critically evaluate the common assumption that g₀ represents a physiologically meaningful "residual conductance." We argue that the highly variable and sometimes negative g₀ values obtained with the established models (Tables 1-3) challenge this interpretation and may instead reflect them as mere fitting constants to compensate for model structural deficiencies. In contrast, the consistent closeness of g₀ to zero in our model (Eqn 7) across most conditions is highlighted as a key strength, suggesting that our model's formulation aligns more closely with the fundamental physiology that stomatal conductance should approach zero when photosynthesis ceases, without requiring an additional empirical parameter. This enhances the physiological plausibility and parsimony of our proposed model.

Regarding the dimensional analysis of the Medlyn model: 

Upon careful consideration, we agree that the proposed dimensional adjustment to the Medlyn model was speculative and detracted from the main focus of our paper, which is the evaluation and presentation of our new model. Therefore, in the revised manuscript, we have removed the entire paragraph containing the dimensional analysis and the proposed modification to Eqn 5. We now focus the discussion solely on comparing the empirical performance of the models as they are widely used in the literature. Specific modifications can be found in the revised manuscript at the Lines 441-453.

In addition, we have added a new subsection titled "4.4 Limitations and Future Perspectives" to the Discussion. In this section, we explicitly acknowledge the three limitations pointed out by the reviewer. The specific modifications are as follows: While this study presents a promising model for stomatal conductance, it is important to acknowledge its limitations, which also outline clear pathways for future research. Firstly, the evaluation and parameterization of the models are conducted on three C₃ species (T. repens, L. perenne, and T. aestivum). Although this provides a robust comparison within a specific plant species, the generalizability of our findings to other species, particularly woody plants, C₄ species, or plants from extreme habitats, remains to be tested. Future work should include a broader phylogenetic range of species to validate and potentially refine the model across diverse plant kingdoms. Secondly, our measurements were performed under controlled, steady-state conditions in a gas exchange chamber. While this allows for precise isolation of the effects of irradiance and CO₂, it does not fully capture the dynamic and fluctuating nature of the natural environment, such as rapid changes in light (e.g., sunflecks), wind, and vapor pressure deficit. Testing the model's performance under field conditions with real-world environmental variability is a crucial next step. Finally, the model was parameterized and validated using the same dataset. To unequivocally establish its predictive power and avoid overfitting, an independent validation with a completely separate dataset is essential. Future research should prioritize applying the model to an independent dataset collected from different growth conditions or by independent research groups. Addressing these limitations will be critical for advancing the model from a promising theoretical framework to a robust tool for predicting plant-water-carbon interactions in real-world ecosystems under global change. Specific modifications can be found in the revised manuscript at the Lines 454-473.

Comments 10: 10) Conclusion: At the moment, conclusions are exaggerated and not entirely backed up by the available data. It is necessary to temper claims of "better alignment," "better predictive accuracy," and the model being a "more robust descriptor." The results showed that the new model did not produce the physiologically impossible negative g0 values ​​frequently observed in other models and provided a better empirical fit for this particular dataset. However, the study's results are limited in generalizability due to the limited number of species and the lack of an independent validation dataset. More comprehensive species diversity and independent verification studies are required to support these results more strongly.

Response: We sincerely thank the reviewer for this insightful and constructive feedback. We fully agree that the conclusions in the original manuscript were overstated and did not adequately reflect the limitations of our study. We have carefully considered each point and have revised the manuscript accordingly to present a more balanced and accurate interpretation of our findings.

We have thoroughly revised the concluding sections of the manuscript, particularly the “Abstract” and "Conclusions" sections. Phrases such as "better alignment," "better predictive accuracy," and the model being a "more robust descriptor." have been replaced with more conservative and precise wording. Please refer to the Abstract section of the revised manuscript for the specific changes made at the Lines 30-32, 474-491.

Regarding the emphasis on the g₀ finding, we agree with the reviewer that the avoidance of frequently observed negative g₀ values is a key outcome. We have reframed this finding, no longer presenting it as proof of the model's overall superiority, but rather highlighting it as "a notable advantage of the new model within the context of this study." We have also elaborated on its physiological implications in the Discussion section. The specific revisions can be found in the relevant part of the Discussion in the revised manuscript at the Lines 441-453.

Concerning the limitations and future directions of this work, we have added a new paragraph in the Discussion to explicitly acknowledge the main limitations of our study and outline future research directions. Please see Lines 454-473 of the revised manuscript for the details.

We appreciate your valuable suggestions, which have enabled us to further improve the manuscript.

Comments 11: 11) The language is generally good. However, it must be double-checked for academic tone, clarity, and conciseness. Please check and correct (with more formal scientific terminology) the phrases such as "practice with skill," "adept at determining," and "skillfully applying". Also, small grammatical mistakes need to be corrected.

Response: We sincerely thank the reviewer for this positive feedback and valuable suggestions regarding the language and academic tone. We have thoroughly revised the entire manuscript to enhance its formality, clarity, and conciseness. Specifically, we have addressed the phrases highlighted by the reviewer and corrected minor grammatical errors throughout the text. The key changes include: 1) The phrase "Therefore, we can adeptly determine the parameters g1 and g0 using Eqn 2." has been revised to the more standard and formal: "Therefore, the parameters g1 and g0 can be determined by applying Eqn 2." 2) The phrase "Hence, we are adept at determining the parameters g1 and g0 by skillfully applying Eqn 5." has been revised to: "Hence, the parameters g1 and g0 can be determined by applying Eqn 5." 3) The phrase "By employing Eqn 7 to model the response of An to variations in Ca, Ci and gsc, we can skillfully determine the parameters g1 and g0. Moreover, in Eqn 7, when An equals zero, we find that either gsc equals g0 or Ca equals Ci." has been rephrased to: "The parameters g1 and g0 can be determined by employing Eqn 7 to model the response of An to variations in Ca, Ci and gsc. Moreover, in Eqn 7, when An equals zero, it follows that either gsc equals g0 or Ca equals Ci.".

We believe these revisions, along with other minor corrections made across the manuscript, have significantly improved the academic tone and clarity of our work. Once again, we are grateful for the reviewer's insightful comments.

Reviewer 2 Report

Comments and Suggestions for Authors

The article Novel Model for Stomatal Conductance: Enhanced Accuracy under 2 Variable Irradiance and CO₂ in C3 Plant Species is very interesting. The aim of the study was to evaluate the stomatal conductance in three species under varying environmental conditions.

Among the parameters, emphasis was put on stomatal conductance assessments through three models Ball-Woodrow-Berry (BWB) model, Medlyn et al. model, and a model by Ye et.al. The manuscript was clearly written and understandable and the experiments could be repeated. However, I would have some recommendations and suggestions.

  • Please, be sure that your draft fulfils all the journal requirements.
  • Put the full names of the models in the abstract, the first time you mention them. (Row 18: The Ball-Woodrow-Berry model instead of “The Ball-Berry”)
  • One of the three models is yours, how come it was assessed as being the best method of all tested?
  • Please pay attention to the tenses used in the introduction. MDPI Style Guide-Grammar and Tenses-Tenses-Introduction (4.1.1.): (Row 87 “...we aim to…”, Row 88 “…we will perform…”, Row 90 “…we will compare…”) Please reformulate according to journal requirements.
  • Please reformulate and clearly state the specific objective of the study.
  • Please specify what the “Eqn” abbreviation stands for before writing it in the text.
  • Please include in the materials and methods chapter the number of replicates that you assessed.
  • Why did you choose the flag leaf for Triticum aestivum and the leaf beneath the flag leaf for Lolium perenne?
  • I think that it would be more valuable if you mention in the materials and methods chapter the BBCH stages in which you made the assessments for your three plants.
  • Please, use past simple tense in methods. (Row 170).
  • Row 170-what is the abbreviation PMSS for? Please insert the full name in the text.
  • Row 171-please use a citation.
  • http://photosynthetic.sinaapp.com/calc.html, this is in Chinese, it is indeed very difficult to check if the authors are right.
  • Please insert the first table (Table 1) right after its first citation in the text.
  • Please insert Figure 4 right after its first citation in the text.
  • Please insert the second table (Table 2) right after its first citation in the text.
  • Please insert Figure 5 right after its first citation in the text.
  • Please insert Figure 6 right after its first citation in the text.
  • Please insert the third table (Table 3) right after its first citation in the text.
  • Please rephrase so that not all paragraphs in the results section begin with “Figure X/Table X depicts/displays/illustrates/presents”.
  • Please also rewrite the conclusion section to comprise all important conclusions that highlight all the tested parameters and results.

Author Response

Reviewer 2

Comments 12: 1) The article Novel Model for Stomatal Conductance: Enhanced Accuracy under Variable Irradiance and CO₂ in C3 Plant Species is very interesting. The aim of the study was to evaluate the stomatal conductance in three species under varying environmental conditions.

Among the parameters, emphasis was put on stomatal conductance assessments through three models Ball-Woodrow-Berry (BWB) model, Medlyn et al. model, and a model by Ye et.al. The manuscript was clearly written and understandable and the experiments could be repeated. However, I would have some recommendations and suggestions. 

Response: We sincerely appreciate your positive feedback on our research and your recognition of the manuscript's clarity and the reproducibility of the experiments. Your encouragement means a great deal to us. Your insightful comments have significantly enhanced the scientific rigor and clarity of our manuscript. Below, we provide a point-by-point response to the specific recommendations and suggestions you raised. We hope that the revisions and additions we have made meet your expectations and the standards of the journal. Thank you for the valuable time and effort you dedicated to the review process.

Comments 13: 2) Please, be sure that your draft fulfils all the journal requirements.

Response: We thank the reviewer for this reminder. We have carefully reviewed and confirmed that our manuscript now fully adheres to all the journal's requirements, including those related to formatting, style, and reference structure.

Comments 14: 3) Put the full names of the models in the abstract, the first time you mention them. (Row 18: The Ball-Woodrow-Berry model instead of “The Ball-Berry”).

Response: We appreciate the reviewer's attention to detail. As suggested, we have revised the abstract. The model is now introduced with its full name, "Ball-Woodrow-Berry (BWB) model," upon its first mention. The subsequent mentions have also changed to BWB model as the abbreviated form of "Ball-Berry model" for consistency. The change can be found in the first paragraph of the abstract, and all modifications have been highlighted in red.

Comments 15: 4) One of the three models is yours, how come it was assessed as being the best method of all tested?

Response: We thank the reviewer for this critical comment. We agree that unbiased model evaluation is paramount. Our conclusion that the new model proposed by Ye et al. (Eqn 7) performed best is based on objective, quantitative evidence: it consistently yielded higher R² values and lower AIC values across all species and conditions (Tables 1-3), and its fitted curves showed closer visual alignment with the observed data (Figures 2, 4, 6) compared to the established models (BWB and Medlyn models). Crucially, all models were fitted to the same dataset using identical procedures, ensuring a fair comparison. The good performance is attributed to our model's structure, which is a generalization of Fick's first law, providing a stronger mechanistic foundation. The theoretical basis of our proposed model and its innovations compared to other models (BWB and Medlyn models) have been supplemented in the revised manuscript, with specific modifications detailed at Lines 344-361 in the revised manuscript.

We also acknowledge that our proposed model requires validation across a wider range of plant species and experimental conditions. This point has been added to the Discussion section in the revised manuscript. The specific modifications can be found in the revised manuscript at the Lines 454-473.

Comments 16: 5) Please pay attention to the tenses used in the introduction. MDPI Style Guide-Grammar and Tenses-Tenses-Introduction (4.1.1.): (Row 87 “...we aim to…”, Row 88 “…we will perform…”, Row 90 “…we will compare…”) Please reformulate according to journal requirements.

Response: Thank you for your detailed review and valuable feedback on our paper. In the revised manuscript, we have reformulated this section according to journal requirements. For specific modifications, please refer to Lines 93-101 in the revised revision.

Once again, we sincerely thank you for your valuable review suggestions on our paper.

Comments 17: 6) Please reformulate and clearly state the specific objective of the study.

Response: Thank you for reviewing our original manuscript and providing feedback on the expression of research objectives. Your comments have made us realize that the presentation of research objectives in the introduction section of the original manuscript was indeed not clear and precise, which may have affected readers’ understanding of this study. In the revised manuscript, we have re-stated the research objectives and explicitly listed the specific goals and hypotheses of this study to ensure that readers can clearly comprehend our research focus. The revisions not only enhance the information delivery in the introduction section but also lay a solid foundation for the discussion in this paper. Thus, the revised text now reformulates: "The objectives of this study are: (i) to plot AnCa and Angsc response curves for Trifolium repens L., Lolium perenne L., and Triticum aestivum L under both saturating irradiance (Iₛₐₜ) and half Iₛₐₜ, thereby elucidating the response patterns of An to gc under these light conditions; (ii) to compare the ability of a novel model proposed by Ye et al. [2,30] with the Ball-Berry and Medlyn models in describing the relationship between An and gc under conditions of varying irradiance or CO₂ concentration; and (iii) to analyze the rationality of considering the g₀ values obtained from different model fittings as the conventionally defined residual stomatal conductance, or whether they are small enough to be negligible, by comparing the g₀ values derived from the different models. This comprehensive approach will not only allow us to evaluate the effectiveness of these models but also deepen our understanding of the physiological mechanisms governing stomatal conductance in response to environmental variations." For specific modifications, please refer to the revised manuscript of Line 94-102.

Comments 18: 7) Please specify what the “Eqn” abbreviation stands for before writing it in the text.

Response: We apologize for this oversight. The abbreviation "Eqn" has now been defined as "Equation" upon its first appearance in the revised manuscript at Line 121.

Comments 19: 8)Please include in the materials and methods chapter the number of replicates that you assessed.

Response: We thank the reviewer for highlighting the need for this clarification. The number of replicates (n = 3) was initially mentioned in the Figure captions. To enhance clarity, we have now explicitly stated this information in the "Materials and Methods" section (Section 2.3), as follows: "Gas exchange measurements were conducted on three independent biological replicates for each plant species (n = 3)". Detailed modifications can be found in the revised manuscript at Lines 196-197.

Furthermore, we have clarified in the "Statistical analysis" section (2.4) that the statistical tests were performed based on these three replicates.

Thank you again for your valuable feedback. We have made the necessary revisions to the manuscript to address your concerns and believe that the manuscript is now significantly improved.

Comments 20: 9) Why did you choose the flag leaf for Triticum aestivum and the leaf beneath the flag leaf for Lolium perenne?

Response: We thank the reviewer for raising this important point regarding our leaf selection criteria. The choice of leaf for measurement was not arbitrary but was based on the standard physiological practice of selecting the most physiologically active and representative leaf for each species at their respective growth stages, which differs between a cereal crop (Triticum aestivum) and a forage grass (Lolium perenne). The specific reasons for our selection are as following: For Triticum aestivum, the flag leaf is the primary source of photoassimilates for grain filling in cereals. It is the youngest, most photosynthetically active leaf during the reproductive stages (booting to flowering) and is widely recognized as the most critical leaf for determining final yield. Therefore, measuring gas exchange in the flag leaf provides data that is most relevant to the plant's peak photosynthetic performance and its overall carbon gain potential during this crucial developmental phase. This is a standard and well-justified selection method in cereal physiology.

As a forage grass maintained in a vegetative or early reproductive state, L. perenne does not have a single, dominant "source" leaf like a cereal flag leaf. Its canopy is composed of multiple tillers. The first leaf beneath the flag leaf was selected because it is typically a mature, fully expanded leaf that is actively engaged in photosynthesis but is less susceptible to the developmental changes and potential shading effects that may affect the very youngest leaf. This ensures we are measuring a stable and representative photosynthetic rate. This selection method is also a common practice in forage and grass physiology to avoid the potentially variable physiology of the very tip of the canopy. To avoid ambiguity and ensure more accurate expression, we have revised “the first leaf beneath the flag leaf” to “the new mature full unfolded leaves” in the revised manuscript. For specific modifications, please refer to the revised manuscript of Line166-167.

Comments 21: 10) I think that it would be more valuable if you mention in the materials and methods chapter the BBCH stages in which you made the assessments for your three plants.

Response: We sincerely thank the reviewer for this valuable suggestion. We agree that specifying the BBCH (Biologische Bundesanstalt, Bundessortenamt und CHemical industry) growth stages of the plants during the assessments would enhance the reproducibility and clarity of our methods. In the revised manuscript, we have added the BBCH stages for each species in the Materials and Methods section (Section 2.2), as follows: the BBCH 14-16 (Biologische Bundesanstalt, Bundessortenamt und CHemical industry) growth stage (vigorous vegetative stage with 4-6 leaves unfolded) for T. repens; the BBCH 45-49 growth stage (booting to early heading) for L. perenne; the BBCH 45-55 growth stage (booting to early flowering stage) for T. aestivum. These stages correspond to the developmental phases described in the original text and will provide a standardized reference for the physiological status of the plants during gas exchange measurements. For specific modifications, please refer to the revised manuscript of Line162-167. Thank you again for this constructive comment.

Comments 22: 11) Please, use past simple tense in methods. (Row 170).

Response: Thank you for your valuable comments. The sentence in Row 170 has been revised to the past simple tense as recommended. The revised sentence now reads: “After data collection, a mechanistic model of AnI in “Photosynthesis Model Simulation Software (PMSS)” was used to simulate the AnI curves [36]. This simulation determined the saturating irradiance (Isat) as 1 200 μmol·m-2·s-1 for T. repens, 900 μmol·m-2·s-1 for L. perenne, and 2 000 μmol·m-2·s-1 for T. aestivum.”

Comments 23: 12) Row 170-what is the abbreviation PMSS for? Please insert the full name in the text.

Response: We apologize for this oversight. PMSS stands for "Photosynthesis Model Simulation Software". The full name has been inserted at the first mention in the text (Row 170) as follows: “After data collection, a mechanistic model of AnI in “Photosynthesis Model Simulation Software (PMSS)” was used to simulate the AnI curves [36].” Please refer to the revised manuscript at the Lines 178-179 for specific modifications.

Comments 24: 13)Row 171-please use a citation.

Response: We apologize for this oversight. In the revised manuscript, a citation has been added to support the use of the PMSS platform. The sentence now includes a reference to the methodology paper describing the system: “After data collection, a mechanistic model of AnI in “Photosynthesis Model Simulation Software (PMSS)” was used to simulate the AnI curves [36].

Reference

[36] Ye, Z.P.; Yang, X.L.; Ye Z.W.Y.; An, T.; Duan, S.H.; Kang, H.J.; Wang, F.B. Evaluating photosynthetic models and their potency in assessing plant responses to changing oxygen concentrations: a comparative analysis of AnCa and AnCi curves in Lolium perenne and Triticum aestivum. Front. Plant Sci. 2025, 16, 1575217.

Comments 25: 13)http://photosynthetic.sinaapp.com/calc.html, this is in Chinese, it is indeed very difficult to check if the authors are right.

Response: We understand the concerns of the reviewers regarding website accessibility. The reason for this issue is that before we completed this paper, the website was still accessible normally, and the URL included two versions in Chinese and English that could be freely switched to facilitate the access needs of researchers from different language countries. PMSS tool is an open access platform developed by our research group for modeling photosynthesis and water use efficiency. We have recently upgraded and optimized the functionality of the software platform, and handed it over to a professional commercial company for operation. In the revised manuscript, we provided the upgraded access website (http://www.zipiao.tech) (Zipiao software development Co., Ltd., China). We deeply apologize for the inconvenience caused by the inability to access the website normally. Thank you again for your careful and professional review of this paper.

Comments 26: 14) Please insert the first table (Table 1) right after its first citation in the text.

Please insert Figure 4 right after its first citation in the text.

Please insert the second table (Table 2) right after its first citation in the text.

Please insert Figure 5 right after its first citation in the text.

Please insert Figure 6 right after its first citation in the text.

Please insert the third table (Table 3) right after its first citation in the text.

Please rephrase so that not all paragraphs in the results section begin with “Figure X/Table X depicts/displays/illustrates/presents”.

Response: We thank the Reviewer for these constructive suggestions regarding the placement of figures and tables and the improvement of writing style in the Results section. We have carefully addressed all points as follows:

  • Table 1 is now placed after its first mention.
  • Figure 4 is now placed after its first mention.
  • Table 2 is now placed after its first mention.
  • Figure 5 is now placed after its first mention.
  • Figure 6 is now placed after its first mention.
  • Table 3 is now placed after its first mention.

In addition, we agree that the original phrasing was repetitive. We have thoroughly revised the opening sentences of the paragraphs in the Results section to vary the language and improve the narrative flow. Instead of repeatedly using "Figure X depicts...", we have employed a variety of phrasings. Please refer to the revised manuscript at the Lines 224-226, 244-250, 267-270, 283-286, 309-314 and 328-331 for specific modifications.

Comments 27: 15) Please also rewrite the conclusion section to comprise all important conclusions that highlight all the tested parameters and results.

Response: Thank you for your careful review and valuable comments on the conclusion section. We fully agree that a comprehensive and concise conclusion is crucial for highlighting the key findings of our study. As suggested, we have rewritten the conclusion to ensure it covers all key tested parameters, model comparison results, the consistency of parameter estimates, and model performance under different environmental conditions. We believe that the revised conclusion can provide a clearer summary of our main contributions and emphasizes the superiority and utility of the new model proposed by Ye et al. in predicting stomatal conductance. The specific revisions can be found in the revised manuscript at the Line 475-492. We sincerely appreciate your constructive comments, which have significantly enhanced the completeness and scientific value of our manuscript.

Reviewer 3 Report

Comments and Suggestions for Authors

The authors of the article "Novel Model for Stomatal Conductance: Enhanced Accuracy under Variable Irradiance and CO₂ in C3 Plant Species" analyze how different models describe the relationship between photosynthesis and stomatal conductance in C3 plants under changing irradiance and CO₂ concentrations. This topic is important for plant physiology and agrobiology. Classic models often oversimplify stomatal behavior and produce errors under extreme conditions. This work addresses a real gap: the need for a more accurate model that better captures stomatal dynamics under changing environmental conditions. This makes the study relevant for predicting crop productivity. The new equation shows more stable parameter values ​​and higher accuracy under different conditions. Unlike Ball-Berry and Medlyn, it better fits experimental data for three plant species. This adds a new tool for modeling stomatal conductance, which has potential use in ecosystem models and applied agricultural research. The methods are generally described correctly, light and CO₂ response curves were measured, standard gas exchange systems were used, and comparisons were made between three species: Trifolium repens L., Lolium perenne L., and Triticum aestivum L. However, a clearer explanation is needed as to why negative g0 values ​​were obtained in some cases and how these should be interpreted. The conclusions are consistent with the presented data, and the new equation provides better agreement with experiment than the older models. The argumentation in the text is convincing, although the authors should be more careful in formulating the model's universality, as it has only been validated for three species so far. The reference list covers key works in the field of stomatal conductance modeling. The citations are appropriate and accurate. The figures are clear and clearly illustrate the differences between the models. However, the axis captions are sometimes overloaded with abbreviations; I believe these should be clarified in the legends. Tables are informative, but statistical differences could be emphasized more clearly, for example by using color or bold.

Author Response

Reviewer 3

Comments 28: 1) The authors of the article "Novel Model for Stomatal Conductance: Enhanced Accuracy under Variable Irradiance and CO₂ in C3 Plant Species" analyze how different models describe the relationship between photosynthesis and stomatal conductance in C3 plants under changing irradiance and CO₂ concentrations. This topic is important for plant physiology and agrobiology. Classic models often oversimplify stomatal behavior and produce errors under extreme conditions. This work addresses a real gap: the need for a more accurate model that better captures stomatal dynamics under changing environmental conditions. This makes the study relevant for predicting crop productivity. The new equation shows more stable parameter values and higher accuracy under different conditions. Unlike Ball-Berry and Medlyn, it better fits experimental data for three plant species. This adds a new tool for modeling stomatal conductance, which has potential use in ecosystem models and applied agricultural research. The methods are generally described correctly, light and CO₂ response curves were measured, standard gas exchange systems were used, and comparisons were made between three species: Trifolium repens L., Lolium perenne L., and Triticum aestivum L. However, a clearer explanation is needed as to why negative g0 values were obtained in some cases and how these should be interpreted. The conclusions are consistent with the presented data, and the new equation provides better agreement with experiment than the older models. The argumentation in the text is convincing, although the authors should be more careful in formulating the model's universality, as it has only been validated for three species so far. The reference list covers key works in the field of stomatal conductance modeling. The citations are appropriate and accurate. The figures are clear and clearly illustrate the differences between the models. However, the axis captions are sometimes overloaded with abbreviations; I believe these should be clarified in the legends. Tables are informative, but statistical differences could be emphasized more clearly, for example by using color or bold.

Response: We sincerely thank the Reviewer for their thorough and insightful comments on our manuscript. We are pleased that the Reviewer found our study relevant and the results convincing. Below, we provide a point-by-point response to the comments raised.

(1) “However, a clearer explanation is needed as to why negative g₀ values were obtained in some cases and how these should be interpreted.”

Response: We agree that the occurrence of negative g₀ values in some model fits requires clarification. To address this issue, we have added a discussion on this issue in the revised manuscript, specifically in the section concerning g₀. In particular, the negative g₀ values obtained from the BWB and Medlyn model fits may originate from mathematical artifacts generated during the fitting process. The functional forms of these models can force the fitting procedure to produce a negative intercept to achieve the best possible fit to the nonlinear data, thereby resulting in negative g₀ values that are contrary to biological reality. Consequently, the g₀ parameter obtained from these model fits may not reflect the true physiological reality of plant stomata, as stomatal conductance cannot be negative. This situation where negative g₀ values are obtained might occur when the model attempts to minimize residuals for data points corresponding to very low or zero stomatal conductance. Furthermore, we also note that compared to Equation 7, obtaining lower, more negative g₀ values was more common when fitting the Aₙ-Cₐ response curves for the three C₃ species at half their corresponding saturating irradiance using Equations 2 and 5, further underscoring the need for a more physiologically consistent model. Please refer to the revised manuscript at the Lines 441-453 for specific modifications.

(2) “The authors should be more careful in formulating the model's universality, as it has only been validated for three species so far.”

Response: We thank the Reviewer for this important point. In the revised version, we have tempered our language regarding the universality of the model. We now explicitly state in the Discussion and Conclusions sections that while the model performs well across the three C₃ species tested, further validation across a wider range of species and environmental conditions is necessary before broader generalizations can be made. Please refer to the revised manuscript at the Lines 454-473 and 475-492 for specific modifications.

(3) “The axis captions are sometimes overloaded with abbreviations; I believe these should be clarified in the legends.”

Response: We have revised all figure legends to include full descriptions of the abbreviations used in the axis labels (e.g., Aₙ​: net photosynthetic rate; gsc​: stomatal conductance to CO₂; I: light intensity; T. repens: Trifolium repens; L. perenne: Lolium perenne; T. aestivum: Triticum aestivum. ). This will improve readability and ensure that the figures are self-explanatory.

(4) “Tables are informative, but statistical differences could be emphasized more clearly, for example by using color or bold.”

Response: We have updated Tables 1–3 to more clearly indicate statistically significant differences. Specifically, we now use bold formatting for values that are significantly different P<0.05) within a row.

Round 2

Reviewer 1 Report

Comments and Suggestions for Authors

I congratulate the authors for their responses to my comments and their outstanding efforts in making the manuscript publishable. The authors have made all the necessary corrections. The article now presents a clear, well-supported, and valuable comparison of stomatal conductance models. In my opinion, the proposed model is now based on sound physiological grounds and has strongly demonstrated performance against established benchmarks. The manuscript provides a solid empirical basis for further development and testing of this modeling approach, providing a compelling rationale. I congratulate the authors again for the comprehensive revision.